# Multimodal Emotion Recognition Calibration in Conversations

## ABSTRACT

Multimodal Emotion Recognition in Conversations (MERC) aims to identify the emotions conveyed by each utterance in a conversational video. Current efforts focus on modeling speaker-sensitive context dependencies and multimodal fusion. Despite the progress, the reliability of MERC methods remains largely unexplored. Extensive empirical studies reveal that current methods suffer from unreliable predictive confidence. Specifically, in some cases, the confidence estimated by these models increases when a modality or specific contextual cues are corrupted, defining these as uncertain samples. This contradicts the foundational principle in informatics, namely, the elimination of uncertainty. Based on this, we propose a novel calibration framework CMERC to calibrate MERC models without altering the model structure. It integrates curriculum learning to guide the model in progressively learning more uncertain samples; hybrid supervised contrastive learning to refine utterance representations, by pulling uncertain samples and others apart; and confidence constraint to penalize the model on uncertain samples. Experimental results on two datasets show that the CMERC significantly enhances the reliability and generalization capabilities of various MERC models, surpassing the state-of-the-art methods.

## CCS CONCEPTS

• **Information systems** → **Multimedia information systems**; **Sentiment analysis**; **Clustering and classification**; • **Computing methodologies** → *Discourse, dialogue and pragmatics*.

## KEYWORDS

Multimodal Conversational Emotion Recognition, Confidence Calibration, Curriculum Learning, Contrastive Learning

## 1 INTRODUCTION

Emotion Recognition in Conversations (ERC) is a challenging task due to the dynamic and spontaneous nature of conversations, where individuals express various emotions [72]. Traditional ERC paradigms rely solely on text [71], but textual cues are often insufficient for understanding deep emotions [17]. Multimodal ERC (MERC), incorporating audio and visual cues alongside the text, is gaining increasing research attention [67].

Current MERC research focuses on two aspects: Firstly, exploring speaker-sensitive context dependencies using recurrent-based network [11, 38], transformer-based network [32, 72], and graph-based

Permission to make digital or hard copies of all or part of this work for personal or classroom use is granted without fee provided that copies are not made or distributed for profit or commercial advantage and that copies bear this notice and the full citation on the first page. Copyrights for components of this work owned by others than the author(s) must be honored. Abstracting with credit is permitted. To copy otherwise, or republish, to post on servers or to redistribute to lists, requires prior specific permission and/or a fee. Request permissions from permissions@acm.org.

*ACM MM, 2024, Melbourne, Australia*

© 2024 Copyright held by the owner/author(s). Publication rights licensed to ACM.
ACM ISBN 978-x-xxxx-xxxx-x/YY/MM
https://doi.org/10.1145/nnnnnnn.nnnnnnn

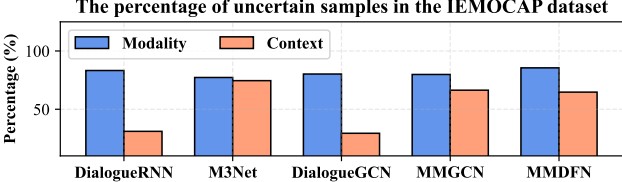

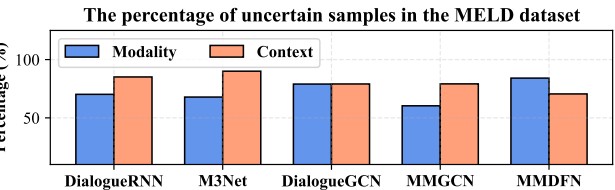

**Figure 1: Percentage of uncertain samples in the testing sets of two datasets stems from removed modalities or contexts.**

network [44, 71]. Secondly, there is significant attention on the fusion of multimodal data, including aggregation-based methods like concatenation [17, 60], tensor product [35, 37], and attention networks [53, 67], as well as heterogeneous graph methods [4, 20, 22].

Despite the above progress, the reliability of current MERC methods remains largely unexplored. In classification settings, a crucial aspect of reliability involves developing a robust confidence estimator [6, 36, 41] that accurately quantifies the probability of correct predictions. Such an estimator proves particularly valuable in high-stakes situations [15]. In MERC, alongside the precise overall prediction confidence, it is crucial to consider the correlation between confidence and modalities or contextual cues, both intra-speaker (within the same speaker) and inter-speaker (between different speakers) contexts. Intuitively, the confidence of the target category should not increase when a modality or specific contextual cues (intra- or inter-speaker contexts) are removed in a MERC model, as the observed information becomes less comprehensive.

However, empirical studies on existing methods reveal a counterintuitive trend – in some cases, the removal of certain modalities or specific contextual cues can lead to an increase in confidence as illustrated in Fig. 1, defining these as uncertain samples. Especially, in multi-party conversations like MELD [47], removing context poses a greater risk of unreliable predictions for MERC models compared to removing modality. Conversely, in dyadic conversations such as IEMOCAP [3], removing modality is more likely to result in unreliable predictions than removing context. This contradicts the fundamental principle in informatics that "the essence of information is to eliminate uncertainty" [2]. This further hampers the reliability of models, making them susceptible to influence when a modality or specific contextual cues are corrupted, as they lack a trustworthy confidence estimator for decision-making. To delve into the underlying reasons, the advanced model M3Net [4] is examined as an example. The analysis reveals that its contradiction with the fundamental principle in informatics arises from its

excessive focus on the textual modality and its struggle to balance the impact of different contexts. Notably, this contradiction seldom results from information removal as noise.

To address the above issue, potential solutions such as temperature scaling [12], Bayesian learning [5, 25], etc., offer global calibration of predicted confidences. However, these methods cannot explicitly calibrate MERC models across various modalities and contexts. Therefore, we propose the CMERC, a framework for calibrating them without altering the model structure, which explores three pivotal calibrations. **(1) Calibrating the training strategy:** Intuitively, the MERC model struggles to make decisions based on reliable predictive confidence for these uncertain samples, which hampers the learning process. Our CMERC employs Curriculum Learning (CL) [1] to guide the model in progressively learning more uncertain samples that contradict the foundational principle in informatics, i.e., the removal of a modality or specific contextual cues enhances the model's confidence, thus aiding in the model's learning. **(2) Calibrating utterance representations:** If the MERC model fails to learn essential features relevant to violations of fundamental informatics principles, reliable confidence estimation becomes difficult. Understanding the factors behind uncertain samples is vital for improving training reliability. We define uncertain samples as positive and others as negative. Hybrid Supervised Contrastive Learning (SCL) is utilized to distinguish uncertain samples from others, facilitating the model in capturing the correlation and difference between uncertain samples and the rest. This process allows the model to identify factors contributing to uncertainty. **(3) Calibrating the loss function:** Introducing a Confidence Constraint aims to constrain unexpected surges in predicted confidences directly. The main contributions of this paper can be summarized as follows:

- We have conducted extensive empirical studies revealing that existing MERC methods fail to provide reliable confidence estimation for decision-making.
- We propose the CMERC framework to calibrate MERC models for the first time, focusing on calibrating the training strategy, utterance representations, and the loss function.
- Experimental results demonstrate that our CMERC can significantly enhance various baselines in generalization and reliability, surpassing the state-of-the-art MERC methods.

## 2 RELATED WORK

**Context modeling in ERC:** Contextual information in conversations provides significant clues for emotion analysis [59]. According to emotion dynamics in conversations [10], the ERC model requires modeling both context- and speaker-sensitive dependencies [57], including recurrent-based network [21, 29, 38], transformer-based network [24, 32, 51], and graph-based network [11, 52, 58]. However, modeling contextual interactions among different modalities remains a significant challenge. Recent research efforts [22, 31] have explored the modeling of intra- and cross-modal interactions within a graph framework to capture contextual clues. Nevertheless, textual cues prove insufficient for understanding deep emotions [17].

**Multimodal fusion:** As multi-modality draws nearer to real-world application scenarios, MERC has been garnering growing research attention in recent years [53]. It integrates modalities like audio and visual cues alongside text to better grasp conveyed emotions [65]. Multimodal fusion in MERC aims to combine information from different modalities, including aggregation-based methods like concatenation [17, 60], tensor product [35, 37], attention networks [49, 62]. However, aggregation-based fusion methods overlook the complex interactions between modalities, resulting in insufficient utilization of contextual information [20]. Recently, researchers have explored graph-based fusion methods to capture intra- and inter-modal interactive information [4, 20, 22, 44, 71]. For instance, Hu et al. [22] investigated intra- and cross-modal interactions in graph networks for contextual clue capture. Despite the progress, the reliability of these methods remains largely unexplored.

**Uncertainty estimation:** Uncertainty estimation is crucial for reliable predictions [36]. Various models have been proposed to address uncertainty, including Bayesian neural networks [7, 26], Dropout [40], and Deep ensembles [16, 28]. In classification tasks, prediction confidence is essential, but models often exhibit overconfidence due to the rapid growth of softmax probabilities [19]. To mitigate this, methods have been developed to calibrate confidence scores to reflect predictive uncertainty [39]. Some approaches aim to train well-calibrated models directly [18, 30, 33, 64, 66, 69, 73], employing techniques like mixup [56], label smoothing [43], and focal loss [41, 42, 74]. Others rely on post-processing methods for calibration [14, 27, 46, 48], with temperature scaling [12] being a prominent example, adjusting probabilities using a single scalar parameter. However, these methods fail to consider the relationship between various modalities or contexts, solely adjusting overall confidence without specific calibration for individual modalities or contexts. This limits their effectiveness in the MERC task.

## 3 METHODOLOGY

In this section, we offer a comprehensive introduction to each component of the CMERC framework as illustrated in Fig. 2.

### 3.1 Task Definition

Let $\mathbf{U} = [\mathbf{u}_1, ..., \mathbf{u}_N]$ be a conversation uttered by $\mathbf{M} \geq 2$ speakers, consisting of N utterances. Each utterance $\mathbf{u}_k$ is represented by a triplet $\mathbf{x}_k = \{\mathbf{x}_k^A, \mathbf{x}_k^V, \mathbf{x}_k^T\}$. $\mathbf{x}_k^A \in \mathbb{R}^{d_a}$, $\mathbf{x}_k^V \in \mathbb{R}^{d_v}$, and $\mathbf{x}_k^T \in \mathbb{R}^{d_t}$ denote the acoustic, visual, and textual features of $\mathbf{u}_k$, respectively. MERC aims to predict the emotion label of each utterance $\mathbf{u}_k$ according to its context $\mathbf{c}_k = \{c_k^X, c_k^O\}$. $c_k^X$ and $c_k^O$ are the intra-speaker and inter-speaker contexts of the utterance $\mathbf{u}_k$.

### 3.2 Feature Representation

Following Ghosal et al. [9], we employ layer normalization and average operation on the last four hidden layers of the Roberta model [34] to obtain textual features. For acoustic and visual feature extraction, following Hu et al. [20], Wen et al. [63], we utilize OpenSmile [8] and a pre-trained DenseNet model [23], respectively[1].

### 3.3 Overview

Considering the unreliable emotional inferences in MERC models, we propose a CMERC framework to calibrate them, as shown in Fig. 2, which integrates three key calibrations: First, it employs CL

---

[1]Please refer to the supplementary materials for detailed feature extraction.

Figure 2: The proposed CMERC framework. Mathematical symbols are consistent with the formulas in the paper.

to progressively train the model on uncertain samples, aiding in the model's learning. Secondly, Hybrid SCL pulls uncertain samples and others further apart, reinforcing the model's focus on factors contributing to uncertainty. Finally, introducing a Confidence Constraint to penalize the model on uncertain samples, ensures a trustworthy confidence estimator for decision-making.

## 3.4 Curriculum Learning for MERC

To design the curriculum for MERC models, we gauge the difficulty of each conversation across different modalities and contexts by measuring confidence levels after the removal operation. Intuitively, the MERC model faces challenges in making decisions based on reliable predictive confidence for uncertain samples, thereby hindering the learning process. Greater improvement on uncertain samples within a conversation, the more difficult it becomes, because the model's confidence becomes increasingly unreliable. As the number of uncertain samples in a conversation increases, the model's predictive confidence becomes less reliable, making it more challenging to grasp the emotion of the utterances. Subsequent experiments have also demonstrated that this difficulty may manifest in the increased entropy of the model's predicted distributions.

$$\mathbf{DF}(\mathbf{d}_i) = \frac{\mathbf{dF}_{\mathbb{C}}(\mathbf{d}_i) + \mathbf{dF}_{\mathbb{m}}(\mathbf{d}_i) + \mathbf{N}_s(\mathbf{d}_i)}{\mathbf{N}_u(\mathbf{d}_i) + \mathbf{N}_s(\mathbf{d}_i)} \tag{1}$$

$$\mathbf{dF}_{\mathbb{m}}(\mathbf{d}_i) = \sum_{k=1}^{\mathcal{B}} \sum_{\xi}^{\{A,V,T\}} \mathbf{CF}_k^{\xi} \tag{2}$$

$$\mathbf{dF}_{\mathbb{C}}(\mathbf{d}_i) = \sum_{k=1}^{\mathcal{B}} \sum_{\xi}^{\{X,O\}} \mathbf{CF}_k^{\xi} \tag{3}$$

$$\mathbf{CF}_k^{\xi} = \max(0, \mathbf{o}_k^{\xi}[\varphi] - \mathbf{o}_k[\varphi]) \tag{4}$$

where $\mathbf{CF}_k^{\xi}$ represents the confidence boost when modalities or contexts are removed. $\mathbb{C} = \{X, O\}$ denotes intra- and inter-speaker contexts. $\mathbb{M} = \{A, V, T\}$ denotes the acoustic, visual, and textual

modalities. $\mathbf{dF}_{\mathbb{m}/\mathbb{C}}(.)$ denotes the modality- or context-specific difficulty measurer. $\varphi$ is the index of target categories. $\mathbf{o}_k^{\xi}$ is the predictive distribution of the MERC model $\mathcal{M}$ with a modality or context removed. $\mathcal{B}$ is the size of mini-batch $\mathbf{d}_i$. $\mathbf{N}_u(\mathbf{d}_i)$ represents the total number of utterances in a mini-batch $\mathbf{d}_i$. $\mathbf{N}_s(\mathbf{d}_i)$ is the number of speakers take part in $\mathbf{d}_i$ and it acts as a smoothing factor. We utilize baby step training scheduler [55] to arrange conversations and organize the training process, described as *Lines* 1 - 4 in Algorithm 1.

## 3.5 Hybrid Supervised Contrastive Learning

Understanding the factors contributing to uncertain samples is pivotal for bolstering the reliability of MERC models during training. Consequently, we advocate for the adoption of a Hybrid SCL framework, which seamlessly integrates modality- and context-specific SCL components. This approach serves to discern uncertain samples caused by the removal of modalities or contextual cues from their counterparts, thereby capturing their nuanced correlations and distinctions. As a result, the MERC model can adeptly identify the underlying factors driving uncertainty during the training process, thereby enhancing its overall effectiveness.

$$\mathcal{L}_m = -\frac{1}{\mathcal{B}} \sum_{\xi}^{\{A,V,T\}} \log\left(\Gamma(\mathbf{z}^{\xi})\right) \tag{5}$$

$$\mathcal{L}_c = -\frac{1}{\mathcal{B}} \sum_{\xi}^{\{X,O\}} \log\left(\Gamma(\mathbf{z}^{\xi})\right) \tag{6}$$

$$\Gamma(\triangle) = \frac{\sum_{j=1}^{\mathcal{B}} \mathbb{1}_{[i \neq j]} \mathbb{1}_{[\triangle_i = \triangle_j]} \ell(\mathbf{H}_i, \mathbf{H}_j)}{\sum_{k=1}^{\mathcal{B}} \mathbb{1}_{[i \neq k]} \ell(\mathbf{H}_i, \mathbf{H}_k)} \tag{7}$$

where $\mathbf{H}$ denotes the hidden representation of the model $\mathcal{M}$. $\mathbf{z}$ indicates the set of pseudo labels, generated according to the process described in *Lines* 14 - 17 of Algorithm 1. $\ell(\star, \star) = e^{simi(\star,\star)/\tau}$, where $\tau$ is the temperature parameter. $simi(\star, \star)$ denotes the cosine similarity function. The calculation process of modality- and context-specific SCL, taking audio and intra-speaker context as an example, is detailed in *Lines* 18 - 27 of Algorithm 1.

**Algorithm 1:** Training process of CMERC using audio and intra-speaker context.

**Input:** Dataset $\mathbf{D}$; the number of buckets $\mathbf{N}_d$; the difficulty measurer $\mathbf{DF}(.)$.

**Output:** $\mathcal{L}_m$, $\mathcal{L}_c$, $\mathcal{L}_s$.

1 ▷ Curriculum Learning in the CMERC.
2 $\hat{\mathbf{D}} = \{\hat{\mathcal{D}}_i\}_{i=1}^{\mathbf{N}_d} \leftarrow sort(\mathbf{D}, \mathbf{DF}); \quad \mathcal{D}_{train} = \varnothing$
3 **for** $i = 1$ to $\mathbf{N}_d$ **do**
4 $\quad \mathcal{D}_{train} = \mathcal{D}_{train} \cup \hat{\mathcal{D}}_i$
5 $\quad$ ▷ Traversing $\mathcal{D}_{train}$ with $\mathbf{N}_b$ mini-batches.
6 $\quad$ **for** $d = \{\mathbf{d}_1, ..., \mathbf{d}_{\mathbf{N}_b}\}$ **do**
7 $\quad\quad \{\mathbf{x}_k^{\{A,V,T\}}, \mathbf{c}_k^{\{X,O\}}\}_{k=1}^{\mathcal{B}} \leftarrow \mathbf{d}; \quad \Gamma_s \leftarrow 0$
8 $\quad\quad$ ▷ Hybrid SCL in the CMERC.
9 $\quad\quad \mathbf{z}^A, \mathbf{z}^X \leftarrow \{0, 0\}_{k=1}^{\mathcal{B}}$
10 $\quad\quad$ **for** $k = 1$ to $\mathcal{B}$ **do**
11 $\quad\quad\quad \mathbf{o}_k, \mathbf{H}_k \leftarrow \mathcal{M}(\mathbf{x}_k^{\{A,V,T\}}, \mathbf{c}_k^{\{X,O\}})$
12 $\quad\quad\quad \mathbf{o}_k^A \leftarrow \mathcal{M}(\mathbf{x}_k^{\{[mask],V,T\}}, \mathbf{c}_k^{\{X,O\}})$
13 $\quad\quad\quad \mathbf{o}_k^X \leftarrow \mathcal{M}(\mathbf{x}_k^{\{A,V,T\}}, \mathbf{c}_k^{\{[mask],O\}})$
14 $\quad\quad\quad$ ▷ Pseudo labeling for the utterance $\mathbf{u}_k$.
15 $\quad\quad\quad$ **for** $\xi = \{A, X\}$ **do**
16 $\quad\quad\quad\quad$ **if** $\mathbf{o}_k[\varphi] < \mathbf{o}_k^{\xi}[\varphi]$ **then**
17 $\quad\quad\quad\quad\quad \mathbf{z}_k^{\xi} \leftarrow 1; \quad \Gamma_s \mathrel{+}= \mathbf{o}_k^{\xi}[\varphi] - \mathbf{o}_k[\varphi]$
18 $\quad\quad \Gamma_A, \Gamma_X \leftarrow 0, 0; \quad \ell_{(+)}^A, \ell_{(-)}^A, \ell_{(+)}^X, \ell_{(-)}^X \leftarrow [\,], [\,], [\,], [\,]$
19 $\quad\quad$ **for** $n = 1$ to $\mathcal{B}$ and $n \neq m$ **do**
20 $\quad\quad\quad$ **for** $\xi = \{A, X\}$ **do**
21 $\quad\quad\quad\quad$ **if** $\mathbf{z}_n^{\xi} == \mathbf{z}_m^{\xi}$ **then**
22 $\quad\quad\quad\quad\quad \ell_{(+)}^{\xi} \mathrel{+}= \ell(\mathbf{H}_n, \mathbf{H}_m)$
23 $\quad\quad\quad\quad \ell_{(-)}^{\xi} \mathrel{+}= \ell(\mathbf{H}_n, \mathbf{H}_m); \quad \Gamma_{\xi} \mathrel{+}= \ell_{(+)}^{\xi} / \ell_{(-)}^{\xi}$
24 $\quad\quad$ ▷ Computing the confidence constraint.
25 $\quad\quad \mathcal{L}_s \leftarrow \Gamma_s / \mathcal{B}$
26 $\quad\quad$ ▷ Computing the contrastive losses.
27 $\quad\quad \mathcal{L}_m, \mathcal{L}_c \leftarrow -\Gamma_A / \mathcal{B}, -\Gamma_X / \mathcal{B}$

## 3.6 Confidence Constraint

To enhance the reliability of predicted confidences in MERC models, we utilize the difference in confidence increase after removal operations as the regularization constraint for mini-batch $\mathbf{d}_i$.

$$\mathcal{L}_s = \sum_{k=1}^{\mathcal{B}} \sum_{\xi}^{\{\mathbb{M}, \mathbb{C}\}} \mathbf{CF}_k^{\xi} \qquad (8)$$

## 3.7 Model Training

We jointly train our proposed framework by minimizing the sum of the following four losses.

$$\mathcal{L} = \mathcal{L}_{ce} + \gamma_m \mathcal{L}_m + \gamma_c \mathcal{L}_c + \gamma_s \mathcal{L}_s + \lambda \|\Theta\|^2 \qquad (9)$$

where $\gamma_m$, $\gamma_c$, and $\gamma_s$ are tuned hyperparameters. $\mathcal{L}_{ce}$ denotes the loss function for the MERC task, typically implemented as a cross-entropy loss. $\Theta$ is the set of trainable parameters within the CMERC. $\lambda$ is the coefficient of $\mathbf{L}_2$-regularization.

# 4 EXPERIMENTS

## 4.1 Datasets

We evaluate the CMERC on two datasets: **IEMOCAP** [3] has dyadic conversation videos with ten speakers, featuring 7,433 utterances and 151 dialogues. Each utterance has one of six emotions. **MELD** [47] contains multiparty conversations collected from the 'Friends' TV series, having 1,433 conversations, 13,708 utterances, and 304 speakers. Each utterance holds one of seven emotions. Following Ghosal et al. [9], the data splitting for datasets is detailed in Table 1. As the IEMOCAP dataset lacks a predefined train/validation split, we allocate 10% of the training dialogues for validation.

## 4.2 Experimental Settings

All re-implementation methods have released their source codes, ensuring identical settings as the original papers. For the CMERC, $\gamma_m$, $\gamma_c$, $\gamma_s$, and $\tau$ are manually tuned for each dataset using hold-out validation[2]. We adopt M3Net [4] as the **Baseline** in this paper. The reported results are the average score of 5 random runs on the test set. Our experiments are conducted on a single RTX 4090 GPU.
**Evaluation metric:** Following Zhang and Li [71], we utilize the weighted F1 score (w-F1) as evaluation metrics and we also report F1 scores per class. To evaluate model prediction reliability, aside from various typically used confidence estimation metrics such as Expected Calibration Error (ECE), Maximum Calibration Error (MCE), Root Mean Square Calibration Error (RMSCE) [13], Area Under the Receiver Operating Characteristic Curve (AUROC), and Area Under the Precision-Recall Curve (AUPRC) [75], we suggest a novel metric called Confidence Enhancement Level (CEL) to measure the degree to which predictive confidence improves for test samples when certain modalities or contexts are removed. A lower CEL denotes more reliable predictions. Except for CEL, all metrics are represented in percentages (%).

$$\mathbf{CEL}(\mathbf{d}_i) = \mathbf{dF}_{\mathbb{C}}(\mathbf{d}_i) + \mathbf{dF}_{\mathbb{m}}(\mathbf{d}_i) \qquad (10)$$

## 4.3 Comparison Models

**Aggregation-based fusion:** Concatenation: DialogueRNN [38] and DialogueGCN [10]; Attention networks: CTNet [70] and SCMM [67]. **Graph-based fusion:** MMDFN [20], MMGCN [22], M3Net [4], CMCF-SRNet [71], and CORECT [44].

Furthermore, we also consider other model-agnostic confidence calibration methods suitable for multi-modal scenarios: T-Scale [12], Ensemble [30], CRL [41], FMFP [74], and CML [36]. These can be seamlessly integrated into the MERC task for a fair comparison.

In our ablation study, we present variations of our proposed CMERC: "**w/o CL**" denotes without CL for MERC; "**w/o HSCL**" denotes without Hybrid SCL (HSCL); "**w/o CC**" denotes without Confidence Constraint (CC).

## 4.4 Reliability Analysis of MERC Models

In Table 2, a significant proportion of uncertain samples is observed in various MERC models, exceeding 90% in some cases, especially when modalities and contexts are removed. Additionally, the impact of each modality or context cannot be underestimated. Interestingly, more advanced models like M3Net show an even higher proportion,

---

[2]Please refer to the supplementary materials for detailed hyperparameter settings.

**Table 1: Statistics of two conversational datasets.**

| Dataset | Dialogues | | | Utterances | | | Classes |
|---|---|---|---|---|---|---|---|
| | train | val | test | train | val | test | |
| MELD | 1039 | 114 | 280 | 9,989 | 1,109 | 2610 | 7 |
| IEMOCAP | 120 | | 31 | 5,810 | | 1,623 | 6 |

**Table 2: Percentage (%) of uncertain samples in the test set. $\mathbb{M}$ represents the removal of any single modality (the union of results from A, V, and T), $\mathbb{C}$ denotes the removal of any single context (the union of results from X and O), and $\mathbb{M} \cup \mathbb{C}$ signifies the union of results from $\mathbb{M}$ and $\mathbb{C}$.**

| Methods | IEMOCAP | | | | | | | |
|---|---|---|---|---|---|---|---|---|
| | A | V | T | | | $\mathbb{M}$ | $\mathbb{C}$ | $\mathbb{M} \cup \mathbb{C}$ |
| DialogueRNN | 43.30 | 45.16 | 56.01 | 12.20 | 27.79 | 83.24 | 31.05 | 88.17 |
| DialogueGCN | 41.47 | 41.10 | 54.16 | 27.05 | 27.85 | 80.22 | 29.39 | 85.52 |
| MMGCN | 36.66 | 35.67 | 53.91 | 30.81 | 49.35 | 79.85 | 66.36 | 95.32 |
| MMDFN | 39.13 | 55.27 | 51.69 | 25.26 | 52.06 | 85.52 | 64.70 | 95.44 |
| M3Net | 36.78 | 47.13 | 33.46 | 35.06 | 49.85 | 77.26 | 74.55 | 91.93 |

| Methods | MELD | | | | | | | |
|---|---|---|---|---|---|---|---|---|
| | A | V | T | X | O | $\mathbb{M}$ | $\mathbb{C}$ | $\mathbb{M} \cup \mathbb{C}$ |
| DialogueRNN | 53.56 | - | 29.96 | 27.89 | 64.56 | 70.19 | 85.10 | 93.49 |
| DialogueGCN | 57.20 | - | 35.86 | 46.01 | 57.24 | 79.00 | 79.08 | 95.17 |
| MMGCN | 33.95 | 26.05 | 28.39 | 51.99 | 55.86 | 60.34 | 79.16 | 92.61 |
| MMDFN | 47.32 | 60.65 | 27.43 | 30.15 | 55.98 | 84.10 | 70.54 | 93.30 |
| M3Net | 30.65 | 42.11 | 26.28 | 60.69 | 46.09 | 67.85 | 90.04 | 97.43 |

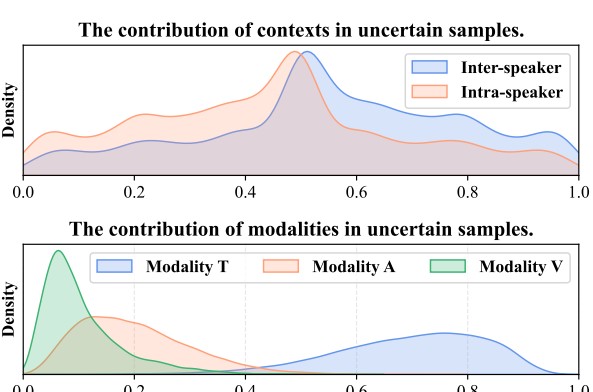

**Figure 3: Contribution of various modalities or contexts in uncertain samples in the IEMOCAP testing set.**

despite enhancements in performance, highlighting the formidable difficulty in reliable confidence prediction for MERC models.

**Further reliability analysis:** We investigate the reasons behind the generation of uncertain samples, focusing on the Baseline as a representative case. In Fig. 3, we visualize the contribution of each modality and context to uncertain samples, as measured by MM-SHAP [45]. This reveals that on the one hand, a significant portion of uncertain samples arises from the model overly prioritizing the textual modality. On the other hand, uncertain samples can also be attributed to the model struggling to appropriately weigh the influence of disparate contexts, treating them indiscriminately.

**Table 3: W-F1 scores (%) of the Baseline[‡] under various missing information rates across different datasets.**

| Missing Rate | 0 | 0.001 | 0.005 | 0.01 | 0.05 | 0.1 | 0.2 | 0.3 | 0.4 |
|---|---|---|---|---|---|---|---|---|---|
| IEMOCAP | 69.61 | 69.45 | 69.26 | 68.57 | 66.83 | 65.79 | 63.17 | 60.46 | 55.09 |
| MELD | 65.37 | 65.09 | 64.88 | 64.55 | 63.82 | 62.55 | 60.36 | 56.64 | 53.24 |

## 4.5 Analysis of Removed Information

Because if the removed information is noise, the increase in predictive confidence is not a bad thing. In fact, the likelihood of increased predictive confidence resulting from noise removal is extremely low for the following reasons: (1) The granularity of removed information is substantial, operating at certain modalities or specific contextual cues rather than at the feature level. The datasets used are of high quality, with almost no instances of pure noise in certain modalities or contexts. (2) The determination of whether removed information is noise depends on the model. As models improve in understanding and denoising, such cases are expected to decrease, particularly as we explore advanced MERC models. (3) Assuming removed information is noise is probable, thus removing it with extremely low probability ideally shouldn't decrease model performance. However, the Baseline in Table 3 shows performance degradation even with a missing information rate as low as 0.001. This highlights that this assumption doesn't hold under such coarse-grained information removal operations.

## 4.6 Overall Results

Table 4 compares our method with others[3], showing its superior W-F1 score and establishing a new state-of-the-art benchmark. Specifically, W-F1 scores rose by 2.37% and 1.48% for IEMOCAP and MELD, respectively. CEL decreased by 629.04 and 194.53 for IEMOCAP and MELD. Table 5 further demonstrates our method's superiority over other confidence calibration methods across various metrics. Importantly, we observed a performance improvement in the advanced MERC model. However, its unreliable confidence estimation has led to deteriorating effects, as evident from the increasing CEL and echoed by other metrics in Table 6. Addressing this issue could further enhance the model's performance, as supported by Table 6. This demonstrates that the CMERC enhances both the reliability and the generalization of various MERC models.

## 4.7 Ablation Study

In this section, we analyze the impact of various components within the CMERC. Ablation experiments in Table 4 show significant improvements across all components. Statistical analysis further confirms this, with a p-value $\ll 0.05$ for the paired t-test.

**Analysis of CL:** The effectiveness of CL is evident in guiding model learning [61], potentially enhancing performance on uncertain samples, as confirmed by ablation results in Table 4. However, the precondition for CL efficacy is an entropy-increasing system [1, 68]. To validate this, we compute the entropy of predicted distributions for uncertain samples and others. In Fig. 4, progressively incorporating uncertain samples contributes to entropy augmentation. Post-CL

---

[3]Please refer to the supplementary materials for a comparison of results against various large language models.

**Table 4: Comparison of results under the multimodal setting. ★ indicates source code available. ‡ denotes our re-implementation results. ♯, ♭, and ♮ represent results come from [20], [54], and original papers, respectively.**

| Methods | IEMOCAP | | | | | | | | MELD | | | | | | | | |
|---|---|---|---|---|---|---|---|---|---|---|---|---|---|---|---|---|---|
| | Happy | Sad | Neutral | Angry | Excited | Frustrated | W-F1 | CEL | Neutral | Surprise | Fear | Sadness | Joy | Disgust | Anger | W-F1 | CEL |
| ★DialogueRNN[b] | 33.67 | 72.91 | 52.32 | 61.40 | 74.24 | 56.54 | 59.75 | **450.94**‡ | 75.50 | 48.81 | 0.00 | 18.24 | 52.04 | 0.00 | 45.77 | 57.11 | 2002.86‡ |
| ★DialogueGCN[♯] | 51.57 | 80.48 | 57.69 | 53.95 | 72.81 | 57.33 | 62.89 | 939.51‡ | 75.97 | 46.05 | - | 19.6 | 51.2 | - | 40.83 | 56.36 | 1730.18‡ |
| CTNet[♮] | 51.30 | 79.90 | 65.80 | 67.20 | 78.70 | 58.80 | 67.50 | - | 77.40 | 52.70 | 10.00 | 32.50 | 56.00 | 11.20 | 44.60 | 60.50 | - |
| ★MMGCN[♯] | 45.14 | 77.16 | 64.36 | 68.82 | 74.71 | 61.40 | 66.26 | 555.78‡ | 76.33 | 48.15 | - | 26.74 | 53.02 | - | 46.09 | 58.31 | **499.89**‡ |
| ★MMDFN[♯] | 42.22 | 78.98 | 66.42 | 69.77 | 75.56 | 66.33 | 68.18 | 766.17‡ | 77.76 | 50.69 | - | 22.93 | 54.78 | - | 47.82 | 59.46 | 1275.21‡ |
| SCMM[♮] | 45.37 | 78.76 | 63.54 | 66.05 | 76.70 | 66.18 | 67.53 | - | - | - | - | - | - | - | - | 59.44 | - |
| ★M3Net[b] | 52.74 | 79.39 | 67.55 | 69.30 | 74.39 | 66.58 | 69.24 | 1772.54‡ | 79.31 | 58.76 | 20.51 | 40.46 | 63.21 | 26.17 | 52.53 | 65.47 | 799.58‡ |
| CMCF-SRNet[♮] | 52.20 | 80.90 | 68.80 | **70.30** | 76.70 | 61.60 | 69.60 | - | - | - | - | - | - | - | - | 62.30 | - |
| CORECT[♮] | 59.30 | 80.53 | 66.94 | 69.59 | 72.69 | **68.50** | 70.02 | - | - | - | - | - | - | - | - | - | - |
| Baseline‡ | 57.05 | 76.70 | 70.55 | 66.08 | 77.37 | 64.37 | 69.61 | 1772.54 | 78.80 | 55.93 | **28.89** | **40.83** | 64.29 | 29.57 | 52.56 | 65.37 | 799.58 |
| w/ CMERC (Ours) | 60.73 | **81.89** | **71.65** | 69.51 | 77.45 | 67.02 | **71.98** | 1143.50 | **80.18** | 60.42 | 24.69 | 40.48 | **65.30** | 32.31 | 54.16 | **66.85** | 605.05 |
| w/o CL | 54.95 | 80.79 | 69.46 | 65.51 | **78.99** | 65.51 | 70.29 | 1447.87 | 79.82 | 60.04 | 22.78 | 35.50 | 63.46 | 29.27 | 55.51 | 66.01 | 659.26 |
| w/o HSCL | 59.28 | 80.35 | 70.11 | 66.02 | 77.50 | 66.05 | 70.67 | 1364.62 | 79.65 | 59.82 | 25.00 | 39.89 | 63.02 | 27.91 | 53.50 | 65.94 | 623.82 |
| w/o CC | **61.82** | 81.78 | 70.88 | 66.48 | 76.28 | 63.22 | 70.46 | 1530.88 | 79.86 | 60.54 | 27.59 | 40.35 | 62.17 | 30.89 | 54.16 | 66.23 | 611.77 |

**Table 5: Comparison of results with other model-agnostic confidence calibration methods under the multimodal setting.**

| Methods | IEMOCAP | | | | | | | MELD | | | | | | |
|---|---|---|---|---|---|---|---|---|---|---|---|---|---|---|
| | W-F1 | CEL | ECE ↓ | MCE ↓ | RMSCE ↓ | AUROC ↑ | AUPRC ↑ | W-F1 | CEL | ECE ↓ | MCE ↓ | RMSCE ↓ | AUROC ↑ | AUPRC ↑ |
| Baseline‡ | 69.61 | 1772.54 | 15.21 | 18.18 | 16.18 | 92.30 | 73.36 | 65.37 | 799.58 | 22.58 | 27.84 | 23.15 | 85.78 | 67.57 |
| w/ ★T-Scale‡ | 70.04 | 1850.75 | 16.33 | 19.17 | 17.16 | 92.25 | 73.70 | 65.64 | 624.69 | 23.68 | 30.54 | 24.95 | 86.79 | 69.11 |
| w/ ★Ensemble‡ | 70.31 | 1579.66 | 14.06 | 16.71 | 14.39 | 92.93 | 75.03 | 65.54 | 707.75 | 22.51 | 31.77 | 23.87 | 87.19 | 69.85 |
| w/ ★CRL‡ | 70.78 | 1670.28 | 14.55 | 16.58 | 14.91 | 92.53 | 74.35 | 65.09 | 671.79 | 23.54 | 33.37 | 24.85 | 86.91 | 69.11 |
| w/ ★FMFP‡ | 63.32 | 2673.93 | 22.53 | 26.27 | 23.77 | 89.15 | 64.65 | 60.80 | 625.76 | 30.03 | 35.60 | 30.40 | 84.13 | 64.13 |
| w/ ★CML‡ | 69.19 | 1324.93 | 13.68 | 16.79 | 14.46 | 92.82 | 75.12 | 66.11 | 637.73 | 22.52 | 30.99 | 23.72 | 87.33 | 70.01 |
| w/ CMERC | **71.98** | **1143.50** | **11.27** | **13.33** | **11.97** | **93.02** | **77.01** | **66.85** | **605.05** | **22.13** | **26.18** | **22.74** | **87.56** | **70.07** |

**Table 6: Performance of various MERC methods based on the CMERC framework for generalizability analysis.**

| Methods | IEMOCAP | | | | | | | | MELD | | | | | | | | |
|---|---|---|---|---|---|---|---|---|---|---|---|---|---|---|---|---|---|
| | Happy | Sad | Neutral | Angry | Excited | Frustrated | W-F1 | CEL | Neutral | Surprise | Fear | Sadness | Joy | Disgust | Anger | W-F1 | CEL |
| DialogueRNN‡ | 30.71 | 83.71 | 53.37 | 62.57 | 68.06 | 56.92 | 60.44 | 450.94 | 76.19 | 47.66 | 0.00 | 23.28 | 51.99 | 0.00 | 42.65 | 57.30 | 2002.86 |
| w/ CMERC | 34.34 | 78.48 | 57.62 | 59.04 | 76.84 | 57.80 | 62.43 | 352.10 | 76.42 | 48.38 | 0.00 | 23.90 | 52.90 | 0.00 | 45.92 | 58.11 | 1758.44 |
| DialogueGCN‡ | 44.88 | 77.97 | 59.56 | 60.62 | 69.69 | 57.18 | 62.46 | 939.51 | 76.19 | 27.50 | 0.00 | 12.04 | 40.32 | 5.71 | 36.86 | 51.82 | 1730.18 |
| w/ CMERC | 35.75 | 79.53 | 59.67 | 60.92 | 75.84 | 59.35 | 63.58 | 825.94 | 74.06 | 45.01 | 0.00 | 23.81 | 52.90 | 0.00 | 39.30 | 55.73 | 1593.99 |
| MMGCN‡ | 41.43 | 77.75 | 60.77 | 70.19 | 74.29 | 63.00 | 65.62 | 555.78 | 75.89 | 43.79 | 0.00 | 21.43 | 54.48 | 0.00 | 46.84 | 57.52 | 499.89 |
| w/ CMERC | 43.08 | 78.63 | 62.29 | 69.44 | 76.32 | 61.68 | 66.24 | 435.84 | 76.89 | 47.20 | 0.00 | 27.44 | 54.34 | 0.00 | 45.59 | 58.67 | 474.27 |
| MMDFN‡ | 43.45 | 80.00 | 63.13 | 71.15 | 74.04 | 66.24 | 67.51 | 766.17 | 76.25 | 48.34 | 0.00 | 24.94 | 52.36 | 0.00 | 46.49 | 58.09 | 1275.21 |
| w/ CMERC | 46.26 | 81.64 | 67.50 | 67.49 | 75.29 | 65.98 | 68.83 | 717.01 | 77.34 | 48.57 | 0.00 | 26.20 | 53.41 | 0.00 | 48.02 | 59.11 | 963.40 |
| M3Net‡ | 57.05 | 76.70 | 70.55 | 66.08 | 77.37 | 64.37 | 69.61 | 1772.54 | 78.80 | 55.93 | 28.89 | 40.83 | 64.29 | 29.57 | 52.56 | 65.37 | 799.58 |
| w/ CMERC | 60.73 | 81.89 | 71.65 | 69.51 | 77.45 | 67.02 | 71.98 | 1143.50 | 80.18 | 60.42 | 24.69 | 40.48 | 65.30 | 32.31 | 54.16 | 66.85 | 605.05 |

| Methods | IEMOCAP | | | | | MELD | | | | |
|---|---|---|---|---|---|---|---|---|---|---|
| | ECE ↓ | MCE ↓ | RMSCE ↓ | AUROC ↑ | AUPRC ↑ | ECE ↓ | MCE ↓ | RMSCE ↓ | AUROC ↑ | AUPRC ↑ |
| DialogueRNN‡ | 40.20 | 77.12 | 44.48 | 49.75 | 18.45 | 36.92 | 48.00 | 38.64 | 49.78 | 28.85 |
| w/ CMERC | 16.71 | 61.78 | 21.45 | 50.80 | 19.97 | 28.32 | 46.08 | 31.83 | 50.27 | 29.24 |
| DialogueGCN‡ | 10.77 | 15.06 | 11.04 | 86.65 | 59.96 | 11.88 | 19.25 | 13.00 | 75.97 | 50.65 |
| w/ CMERC | 4.52 | 14.77 | 7.08 | 88.02 | 63.22 | 8.77 | 14.76 | 10.23 | 78.71 | 56.09 |
| MMGCN‡ | 7.56 | 16.27 | 8.31 | 90.44 | 69.34 | 21.40 | 31.24 | 22.54 | 81.39 | 59.29 |
| w/ CMERC | 4.49 | 13.08 | 5.45 | 91.10 | 70.36 | 16.85 | 20.91 | 17.17 | 82.15 | 60.82 |
| MMDFN‡ | 11.54 | 14.46 | 12.10 | 91.89 | 72.47 | 12.62 | 16.04 | 13.12 | 81.29 | 60.19 |
| w/ CMERC | 6.28 | 10.90 | 6.76 | 92.10 | 73.50 | 6.65 | 10.48 | 7.41 | 81.84 | 60.97 |
| M3Net‡ | 15.21 | 18.18 | 16.18 | 92.30 | 73.36 | 22.58 | 27.84 | 23.15 | 85.78 | 67.57 |
| w/ CMERC | 11.27 | 13.33 | 11.97 | 93.02 | 77.01 | 22.13 | 26.18 | 22.74 | 87.56 | 70.07 |

calibration results in decreased entropy in model predictions for uncertain samples, signifying enhanced comprehension of data distribution and improved classification reliability.

**Analysis of HSCL:** In Table 4, HSCL demonstrates the best overall performance on CEL. In Fig. 5, we conduct t-SNE visualization on the intermediate representations of the Baseline and the Baseline

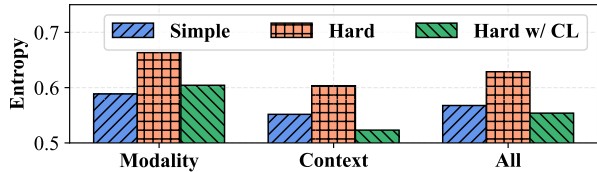

Figure 4: Entropy of the Baseline's predicted distributions on uncertain samples (Hard) and others (Simple).

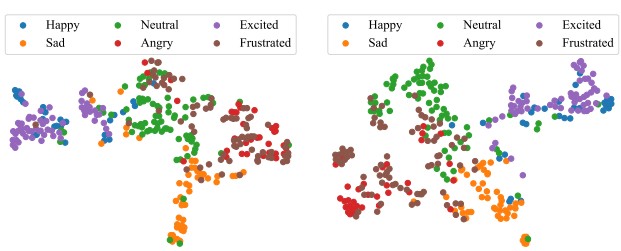

Figure 5: Visualization of intermediate embeddings of uncertain samples from the Baseline (left) and the Baseline w/ HSCL (right) in the IEMOCAP testing set.

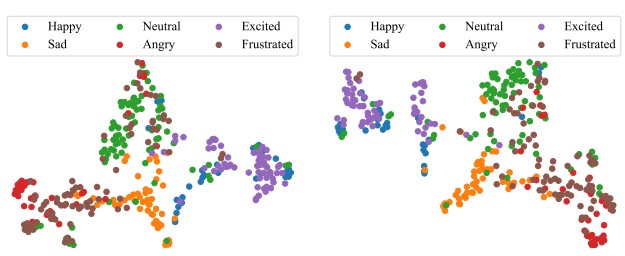

Figure 6: Visualization of intermediate embeddings of uncertain samples from the Baseline w/ Modality-specific SCL (left) and w/ Context-specific SCL (right) in the IEMOCAP testing set.

with HSCL. The latter exhibits significantly clearer distinctions compared to the former, with silhouette coefficients [50] of 0.048 and 0.098, respectively. This suggests that capturing latent features contributing to uncertain samples aids in enhancing utterance representations. In Fig. 6, we also visualize the intermediate representations of the Baseline with modality-specific or context-specific SCL. Their silhouette coefficients of 0.063 and 0.067 surpass those derived from the Baseline, highlighting the efficacy of HSCL and the complementary nature of its constituent components.

**Analysis of CC:** Table 4 shows that CC significantly reduces CEL in the MELD dataset, but it falls short in lengthy conversations like IEMOCAP, possibly due to inefficiencies in the later conversation stages, as supported by Fig. 9. In Fig. 7, considering modalities and contexts in CC leads to CF value (Formula 4) distributions where smaller values correspond to higher density. This underscores CC's effectiveness on uncertain samples and improving model reliability across modalities and contexts in the IEMOCAP testing set.

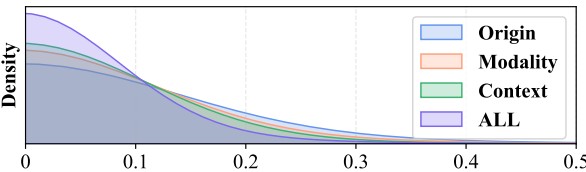

Figure 7: The distribution of CF values under CC.

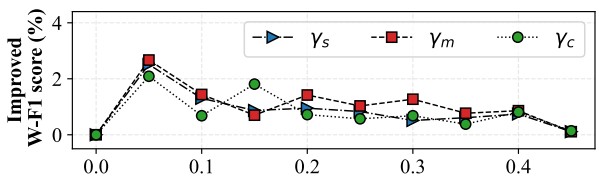

Figure 8: Improved W-F1 score of CMERC across various hyperparameters in the IEMOCAP validation set.

### 4.8 Hyperparameter Analysis

In Fig. 8, we demonstrate improved performance on the IEMO-CAP validation set through the adjustment of hyperparameters, including $\gamma_s$, $\gamma_m$, and $\gamma_c$. These hyperparameters exhibit an initial increase, followed by a decrease and eventual stabilization with minor fluctuations. Importantly, performance consistently surpasses the case where these parameters are set to zero, showcasing the CMERC's effectiveness across different hyperparameter settings.

### 4.9 Comparison under Different Patterns

Our CMERC framework differs from the global calibration of predicted confidences, as it explicitly calibrates the MERC model across various modalities and contexts. In Table 7, we demonstrate the enhanced performance of the CMERC, highlighting substantial advancements in generalization and reliability through individual pattern calibration. This enhancement is particularly noteworthy within the acoustic modality and intra-speaker context. The integration of various modalities yields complementary effects, with consistent findings observed across different contexts.

### 4.10 Complementarity Analysis

In this section, we delve into the complementarity among three pivotal calibrations to elucidate the rationale of the CMERC framework. In Fig. 9, CC performs well in mid-conversation, particularly in the 'Excited' emotion and utterances with ES types, where two consecutive utterances exhibit different emotions. CL demonstrates strength in 'Happy' and 'Angry' emotions, especially in utterances without ES types. HSCL exhibits proficiency in handling 'Neutral' emotions. Collectively, each module showcases strengths in different aspects, underscoring their complementary performance, as also evidenced by Table 4.

### 4.11 Error Analysis

Many errors in our method stem from class imbalance, as evidenced by the low F1 scores of 24.69% and 32.31% for the 'Fear' and 'Disgust'

**Table 7: Analysis of the CMERC across various patterns.**

| Patterns | IEMOCAP | |
|---|---|---|
| | W-F1 | CEL |
| **A / V / T** | 70.41 / 70.39 / 70.32 | 1216.98 / 1293.30 / 1266.92 |
| **A + V** | 70.91 | 1182.20 |
| **A + T** | 70.73 | 1204.88 |
| **V + T** | 70.41 | 1225.06 |
| **A + V + T** | 71.98 | 1143.50 |
| **X** | 71.21 | 1191.55 |
| **O** | 70.68 | 1229.21 |
| **X + O** | 71.98 | 1143.50 |

emotions, respectively, in the MELD dataset. This phenomenon also constitutes a primary constraint on the performance of the MERC task, a fact supported by the results in Table 4. Furthermore, we are also investigating cases where the CMERC-enhanced MERC model misclassifies samples correctly predicted by Baseline[‡], totaling 89 samples in the IEMOCAP dataset. Notably, it struggles particularly in Short (48 samples) and Medium (34 samples) positions of conversations, with relatively better performance observed in Long positions (7 samples). This could be due to the benefits of CL, as CL exhibits outstanding performance in utterances at long positions in conversations, as depicted in Fig 9. Misclassifications are distributed almost evenly across samples with and without ES types. This suggests that our method is not affected by the phenomenon of emotional shift, regardless of whether it is present or not.

### 4.12 Generalizability Analysis

To evaluate the generalizability of our CMERC, we conduct experiments with MERC models, as presented in Table 6. Noticeably, we observe a consistent decrease in CEL and an improvement in W-F1 scores across all methods. Additionally, similar enhancements are reflected in other confidence estimation metrics. These findings demonstrate that the CMERC significantly enhances the reliability and generalization capability of various MERC models.

### 4.13 Case Study

In Fig. 10, we examine misclassified uncertain samples, where color intensity indicates the contribution of modalities or contextual cues. The Baseline tends to favor the textual modality, overlooking other modalities. Through calibration, improvements are observed. For instance, in the first utterance, the model correctly focuses on the visual modality to identify the emotion 'Sad'. Similarly, in the second utterance, emphasizing visual cues helps identify the 'Happy' emotion. For the third and fourth utterances, The Baseline treats inter- and intra-speaker contexts alike, while calibration introduces a preference. Specifically, in the fourth utterance, the model appropriately prioritizes inter-speaker context, recognizing the influence of others on the speaker's 'Frustrated' emotion.

## 5 CONCLUSION

In this paper, we introduce a novel calibration framework CMERC designed to tackle the issue of unreliable predictive confidence in MERC models without altering their structures. It integrates CL to guide the model to learn progressively uncertain samples in an entropy-increasing environment; Hybrid SCL to separate uncertain

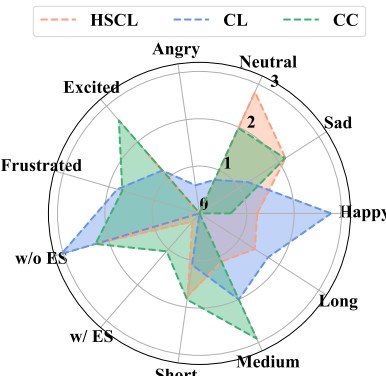

**Figure 9: Improved W-F1 score (%) across various emotional categories, types (ES: Emotion shift), and lengths (Short: first 1/3, Medium: middle 1/3, and Long: last 1/3 of a conversation) in the test set of the IEMOCAP dataset.**

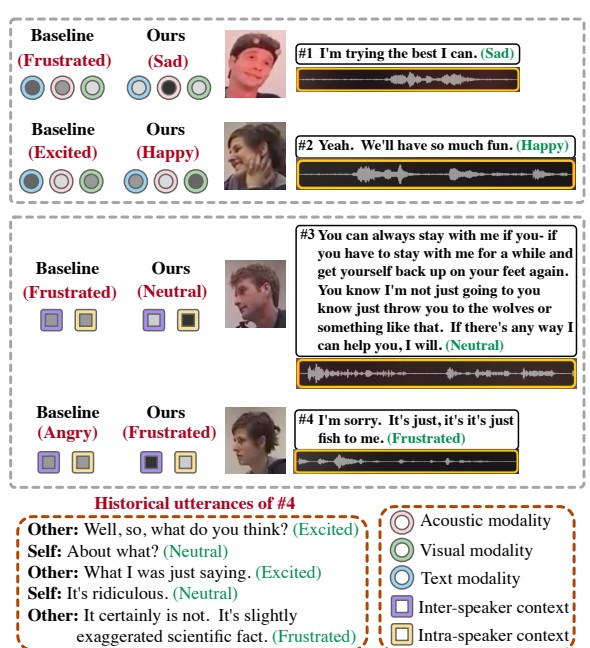

**Figure 10: Examples of utterances in the IEMOCAP testing set for the case study. Predicted and golden labels are highlighted in red and green fonts, respectively.**

samples and reinforce the model's emphasis on factors causing uncertainty; and Confidence Constraint to penalize unexpected confidence surges on uncertain samples. These modules complement each other to improve the reliability of model predictions, particularly on uncertain samples that may arise due to the model's tendency to overly prioritize textual information or struggle to effectively balance different contexts. Experimental results on two datasets show that the CMERC framework significantly enhances the reliability and generalization capability of various MERC models, surpassing state-of-the-art methods.

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
