# OpenReview forum: "Multimodal Emotion Recognition Calibration in Conversations"
_acmmm.org/ACMMM/2024/Conference — MM2024 Poster_

### Official Review · Reviewer_UTji · 2024-05-09

**Rating:** 4
**Confidence:** 3

**Summary:**

This paper introduces a calibration framework for the task of Multimodal Emotion Recognition in Conversations (MERC). The proposed model enhances performance even in scenarios with missing modalities or contextual information.

**Strengths:**

The authors propose a generalizable model for handling incomplete information samples. The experiments are comprehensive and effectively demonstrate the validity of the proposed method. The  figures can help readers to understand this work.

**Limitations:**

The paper delves into many issues related to the MERC task, which may slightly confuse the logical flow. Focusing more on the points described in the abstract would enhance clarity.

The motivation of the paper needs to be reorganized. If the dataset is of the high quality, as mentioned later in the paper, instances of missing modalities or contextual information should be rare. Therefore, the significance of this method in such scenarios needs to be clarified. Validating the method on datasets of lower quality could better illustrate its importance.

Other comments:
1. The introduction's definition of uncertain samples and  Figure 1's results could be smoother. Which rate of missing information leads to the results in Figure 1? Would it be different with different rates?

2. The analysis of the reason for increased confidence after removing some information is not clear. Therefore, the utilization of confidence constraints might be confusing.

3. Are the  hyperparameters in the total loss may make adjustment difficult?

4. Figure 1 in the introduction mainly highlights the differences between two datasets, yet the trends seem consistent in MMDFN. Additionally, there is no significant difference between M3Net and MMGCN. What causes this discrepancy?

P.S. The paper also mentions models dependent on textual modality, an area where much related research has been conducted. Additionally, research on incomplete modality also exists. Exploring these works may be beneficial.

**Suitability:**

3

---

### Official Review · Reviewer_SHeK · 2024-05-24

**Rating:** 2
**Confidence:** 4

**Summary:**

The paper proposes a novel calibration framework CMERC to calibrate MERC models without altering the model structure. CMERC integrates curriculum learning to guide the model in progressively learning more uncertain samples; hybrid supervised contrastive learning to refine utterance representations, by pulling uncertain samples and others apart; and confidence constraint to penalize the model on uncertain samples.

**Strengths:**

The results of the experiment are promising.

**Limitations:**

1. The abstract mentions that the confidence in these model estimates increases when modal or specific context cues are disrupted, thus defining them as uncertain samples. However, what exactly do uncertain samples refer to? Do they refer to missing modalities? I am very confused by the author's expression.
2. To the best of knowledge of my knowledge, existing MERC methods mainly focus on three aspects: speaker-sensitive (graph-based networks), context-sensitive (recurrent-based networks and transformer-based networks), and distinguishing speakers-sensitive (DialogueRNN). The authors need to perform a more detailed classification of existing MERC methods.
3. I am very confused about what Fig. 1 is trying to express. The author does not explain Fig. 1 in detail in either the figure legend or the main text. I don't think Fig. 1 is necessary. Furthermore, the problem defined by the authors is very unclear. I strongly recommend that the authors rewrite the abstract and introduction.
4. How are the uncertain samples in Table 2 counted? Is it the number of misclassifications by the model? In addition, why some data are missing in the video modality condition in the MELD dataset?
5. Why is there no specific meaning given to the abscissa in Fig. 3?
6. The experimental setup of Table 3 is very ambiguous. The author mentions deleting any modality or specific context in the paper, but in Table 3 I am confused that the author does not explain any implementation details of the missing modality.
7. The experimental comparison results in Table 4 are unreasonable. None of the methods compared in the paper are aimed at scenarios where modalities are missing. The authors should compare with some missing modality methods. (e.g., GCNet [1]). Furthermore, why are the missing proportions of the experimental modal and feature contexts not given? I need to know clearly the performance of the model under different missing rates.
8. Since the author did a statistical significance analysis in the ablation experiment, I suggest that the author give the p-values of each emotion category in the supplementary material.
9. The experimental analysis of hyperparameters is relatively rough. The authors should fix one hyperparameter and then use 3D grid search to verify optimal solutions for other parameters.
10. Some academic terminology expressions are not standardized. The tensor product should be a fusion method rather than an aggregation method.
There are some typos errors as follows:
-	L46, P1, Firstly -> firstly
-	L48, P1, network -> networks
[1] Lian Z, Chen L, Sun L, et al. Gcnet: Graph completion network for incomplete multimodal learning in conversation[J]. IEEE Transactions on pattern analysis and machine intelligence, 2023.

**Suitability:**

2

---

### Official Review · Reviewer_A4wV · 2024-05-29

**Rating:** 5
**Confidence:** 3

**Summary:**

The paper proposes CMERC framework to calibrate Multimodal Emotion Recognition models to tackle the challenge of miscalibration in MERC models across various modalities and contexts. The paper performs studies to identify the at existing MERC methods fail to provide reliable confidence estimates. The method utilizes a Hybrid Difficulty Measurer to identify difficult modalities for a given utterance followed by a hybrid supervised contrastive learning to identify factors contributing to uncertain samples. Experiments were conducted on the MELD and IEMOCAP datasets. In addition to the Weighted-F1, the paper presents CEL (Confidence Enhancement Level) to validate the proposed approach.

**Strengths:**

The motivation that each modality contribute differently within an utterance is well-grounded. Identification of miscalibration in existing MERC models provided strong motivation towards presenting confidence estimates in addition to the widely used w-F1 score for multi-class classification datasets including MELD and IEMOCAP. Hence, the proposed Hybrid Difficulty Measurer is a strength of the work.

Extensive ablation studies are provided to validate various components of the proposed method.

Additionally, performance of existing MERC methods based on the proposed CMERC framework is presented to validate generalizability of the proposed work.

**Limitations:**

While the proposed Hybrid Difficulty Measurer sorts the input modalities based on the difficulty score formulated in the paper, the computational complexity of this additional difficulty measurement is not discussed. Additionally, the time taken for training/inference of existing benchmarks (Table 6) with and without CMERC framework would help demonstrate the tradeoff between computational cost v/s performance gains.

**Suitability:**

3

---

### Official Review · Reviewer_fM4X · 2024-06-05

**Rating:** 4
**Confidence:** 2

**Summary:**

The paper proposes a three-pronged approach for improving model reliability for emotion recognition - curriculum learning, contrastive learning and confidence constraints on the loss function

**Strengths:**

- The framework proposed in the paper is very general and can be adopted to any emotion recognition model without any major changes to the architecture
- The paper has extremely robust and varied experiments and ablation studies to support the claims made regarding all three contributions
- The curriculum learning scheme seems to be novel, and provides a good way of measuring difficulty of multimodal samples

**Limitations:**

- The authors should expand on the details behind the difficulty measurement for curriculum learning provided in Sec 3.4, lines 278-284.
- The novelty of the proposed contrastive loss is not clear. While the experiments show that the contrastive loss clearly improves performance, projecting the loss as a primary contribution/novelty of the paper (line 134) does not seem warranted.

**Suitability:**

3

---

### Meta-Review · Area_Chair_j9pt · 2024-07-03

**Recommendation:** Accept (Poster)
**Confidence:** 4

**Metareview:**

Test-Time Adaptation for Multimodal Sentiment Analysis